# Metabolic Plasticity and Epithelial-Mesenchymal Transition

**DOI:** 10.3390/jcm8070967

**Published:** 2019-07-03

**Authors:** Timothy M. Thomson, Cristina Balcells, Marta Cascante

**Affiliations:** 1Department of Cell Biology, Molecular Biology Institute of Barcelona, Science Research Council, 08028 Barcelona, Spain; 2Networked Center for Research in Liver and Digestive Diseases (CIBEREHD), Instituto de Salud Carlos III (ISCIII), 28029 Madrid, Spain; 3Department of Biochemistry and Molecular Biomedicine-Institute of Biomedicine (IBUB), University of Barcelona, 08028 Barcelona, Spain; 4Department of Materials Science and Physical Chemistry, University of Barcelona, 08028 Barcelona, Spain; 5Networked Center for Research in Liver and Digestive Diseases (CIBEREHD) and metabolomics node at INB-Bioinformatics Platform, Instituto de Salud Carlos III (ISCIII), 28029 Madrid, Spain

**Keywords:** epithelial-mesenchymal transition, metabolism, plasticity, epigenetics, drug resistance

## Abstract

A major transcriptional and phenotypic reprogramming event during development is the establishment of the mesodermal layer from the ectoderm through epithelial-mesenchymal transition (EMT). EMT is employed in subsequent developmental events, and also in many physiological and pathological processes, such as the dissemination of cancer cells through metastasis, as a reversible transition between epithelial and mesenchymal states. The remarkable phenotypic remodeling accompanying these transitions is driven by characteristic transcription factors whose activities and/or activation depend upon signaling cues and co-factors, including intermediary metabolites. In this review, we summarize salient metabolic features that enable or instigate these transitions, as well as adaptations undergone by cells to meet the metabolic requirements of their new states, with an emphasis on the roles played by the metabolic regulation of epigenetic modifications, notably methylation and acetylation.

## 1. Introduction

Cells undergoing switches from epithelial to mesenchymal states experience radical changes in motility, proliferation, morphology and interactions with their environment. Epithelial cells that undergo EMT lose cell-cell contacts, undergo extensive cytoskeletal remodeling and exponentially increase their motility and their ability to invade, through extracellular structures as individual cells [1]. At the same time, they adjust their rate of proliferation to the degree of motility, such that highly motile cells with strong acquired mesenchymal phenotypes may exhibit a diminished proliferative potential [2], while cells at “intermediate” states of EMT may retain or increase their proliferation rates relative to their initial epithelial states [3]. This suggests a balance between motility and proliferation [4,5], that may depend on the relative availability of common resources that can be spent on either motility or on proliferation. Indeed, cells can undergo EMT (or also mesenchymal–epithelial transition (MET)) to different extents [6], adopting a range of phenotypes. While “extreme” EMT can lead to stable mesenchymal phenotypes prone to enter pre-senescent states [5], “intermediate” forms of EMT endow cells with features shared with stem cells, including self-renewal or survival under stress and in non-adherent growth conditions [7,8].

Here we will review relevant interconnections between EMT and metabolism, with a particular emphasis on the modulation by metabolites of epigenetic readouts, including EMT. 

## 2. Metabolic Reprogramming in EMT 

As tumor cells proliferate, they require a constant availability of nutrients, and oxygen through cell metabolism for transformation into energy and the molecular components of progeny cells, which include nucleic acids, sugars, proteins, lipids and a myriad of small organic and inorganic molecules. On the other hand, highly motile disseminating tumor cells, or even tumor cells surviving in circulation, prioritize energetic yield over the production of building blocks, to ensure survival and fuel the cellular processes associated to cell motility [1].

While the generation of building blocks is most efficient through precursor, and reducing equivalent production by glycolysis and through precursor intake, the generation of energy in the form of adenosine triphosphate (ATP) is at its most efficient through mitochondrial oxidative phosphorylation (OXPHOS) [9]. However, when certain conditions, such as oncogenic signaling or hypoxia, force growth in spite of an absence of any optimal conditions for mitochondrial function (limited oxygen), cells resort to the activation of mechanisms that favor the derivation of glycolysis towards building block production at the expense of feeding mitochondrial functions (the Warburg effect). 

The metabolic dichotomy established between highly proliferative and highly motile phenotypes is exemplified by a wide variety of observations. For instance, while epithelial cells display fragmented mitochondria dependent on the function of DRP1 mitochondrial fission protein, SNAI1 and TGF-β1-induced mesenchymal cells display mitochondria with predominant mitofusin-dependent fused/tubular morphologies in mammary stem cells [10]. Notably, mitochondrial fusion is associated with enhanced oxidation and tricarboxylic acid (TCA) cycle activity [11,12,13], and then the reversal of mitochondrial fusion leads to a decreased mitochondrial function and a reversal of EMT [10]. In line with this, TGF-β1 induces a shift from glycolysis to OXPHOS by its repression of PDK4, which counters the function of the pyruvate dehydrogenase (PDH) complex to introduce pyruvate into the mitochondria to feed the TCA cycle, as observed in lung cancer cells [14]. Other models of induced EMT, however, appear to favor a diversion of metabolic fluxes away from mitochondria. For example, in basal-like breast cancer, SNAI1 represses fructose-1,6-bisphophatase 1 (FBP1), promoting glucose uptake and the diversion of glycolytic carbons towards biosynthetic pathways, including the pentose phosphate pathway (PPP), and impairing the activity of the respiratory chain complex I [15]. Similarly, SNAI1 also represses phosphofructokinase platelet (PFKP) [16] and several subunits of cytochrome C oxidase (COX) [17] in breast cancer cells, further reinforcing a strongly glycolytic metabolic phenotype. Along the same lines, the silencing of aldolase A (ALDOA) or of glyceraldehyde-3-phosphate dehydrogenase (GAPDH) prevent EMT in various cancer cell models [18,19].

The first challenge that cancer cells face when undergoing the invasion-metastasis cascade resides in acquiring motile and invasive capacities in order to reach the bloodstream. The motile phenotype is driven in part by lipid rafts, cholesterol and sphingolipid-rich membranous structures that modulate cell adhesion by partnering with CD44, and are required for ECM degradation and invadopodia formation [20]. Sphingolipids themselves are also part of other oncogenic signaling cascades that lead to motile phenotypes [21]. Finally, different enzymes in fatty acid metabolism are also recruited for the metastatic process [22]. For example, ATP citrate lyase (ACLY) is required for low molecular weight cyclin E (LMW-E)-mediated transformation, migration, and invasion in breast cancer cells [23]. Similarly, fatty acid synthase (FASN) is also involved in invasion by promoting EMT in breast and ovarian cancer cells [24,25], and by interacting with wingless-related integration site (Wnt) signaling in metastatic colorectal cancer cells [26].

Upon detachment from an adherent layer, healthy non-hematopoietic cells are unable to uptake sufficient glucose, which subsequently leads to ATP shortage, a state that activates anoikis [27,28]. Circulating tumor cells (CTC) display metabolic adaptations specifically devoted to evade anoikis, and to favor anchorage-independent growth, a prerequisite state for metastatic dissemination [29].

Another possible trigger of anoikis is elevated levels of reactive oxygen species (ROS), which also contribute to inhibit ATP production [30]. Therefore, many of the adaptations displayed by CTCs may also be directed to scavenge or diminish the generation of ROS. One of these adaptations may be the reinforcement of a highly-glycolytic phenotype, since relying on glucose to generate ATP can decrease cellular ROS levels by two distinct mechanisms: First, diminished ROS-generating OXPHOS (Crabtree effect; [31]); and, second, increased NADPH production capacity through an enhanced flux through the PPP [32]. In contrast, invasive ovarian cancer cells grown under attachment-free conditions increase the pyruvate uptake for TCA cycle anaplerosis, which favors the adoption of a more oxidative metabolic state and a motile phenotype [33]. Similarly, the colorectal cancer cell line SW620, derived from a lymph node metastasis, exhibits an increase in EMT markers and invasiveness, and an enhanced mitochondrial metabolism to the detriment of aerobic glycolysis, when compared to its primary tumor counterpart, SW480 [34]. 

Not surprisingly, signaling cascades governing metabolic networks and EMT often display a remarkable level of overlapping. A prominent case is illustrated by glycogen synthase kinase-3 β (GSK3β), which targets the EMT transcription factor SNAI1, but is directly phosphorylated and inactivated by Akt. GSK3β phosphorylates SNAI1, targeting it for proteasomal degradation. Thus, Akt activation directly impacts SNAI1 levels through GSK3β, coupling the metabolic and the EMT phenotypes [35]. Importantly, this inhibition of GSK3 activates glycogen synthesis and glucose transport [36]. Thus, the induction of mesenchymal phenotypes by the inhibition of GSK3 and the subsequent SNAI1 (SNAIL) activation is associated with enhanced glucose transport and glycogen accumulation, which may represent a strategy to accumulate carbon and energy reservoirs to sustain cell motility. 

## 3. Metabolism and the Epigenetic Control of Epithelial-Mesenchymal Plasticity

The transitions in EMT and MET are orchestrated by transcription factors, such as ZEB1/2, SNAI1/2, TWIST1/2, and microRNAs, including the miR-200 family and miR-205, whose expression is contingent upon epigenetic states determined by CpG island methylation and histone marks [37]. Enzymes that carry out epigenetic modifications commonly use key metabolites as either substrates or allosteric regulators [38] (Figure 1). Thus, cellular metabolic states can affect epigenetic regulatory proteins through metabolic signaling pathways. Intermediary metabolites involved in epigenetic regulation include acetyl-CoA, S-adenosyl methionine (SAM), α-ketoglutarate (α-KG), ATP, nicotinamide adenine dinucleotide (NAD+) and flavin adenine dinucleotide (FAD). Importantly, the cellular concentrations of many of these substrates can limit enzyme reaction rates [39]. As such, the availability and balance of metabolic resources can significantly modulate chromatin remodeling and transcription factor activities.

### 3.1. Methylation

All methylation reactions require the one-carbon donor SAM, which is synthesized in the methionine cycle from methionine and ATP by methionine adenosyl-transferase (MAT). The transfer of the methyl group from SAM to the substrate produces S-adenosyl homocysteine (SAH), which in turn is converted to homocysteine, and finally regenerated to methionine after the donation of a methyl group from 5-methyl-tetrahydrofolate (5-MTHF) [40]. Methyl moieties are transferred from SAM to acceptor cytosine residues on DNA or lysine and arginine residues on histones in reactions catalyzed by DNA or histone methyltransferases, respectively (Figure 2). The availability of methionine determines the levels of histone and DNA methylation by modulating SAM and SAH [41]. These modifications are sensitive to the abundance of SAM or the [SAM]:[SAH] ratio, since SAH can act as an inhibitor of methyltransferases, and the Km and Ki for SAM and SAH, respectively, fall within physiological ranges [40]. The intracellular methionine concentration is also dependent on the expression of the cell surface amino acid transporter LAT1 (SLC7A5), which forms a heterodimer with SLC3A2 [42], and whose expression is regulated by MYC and the Akt-mTOR pathway [43,44,45].

SAM levels are also dependent on serine metabolism. Serine is a one-carbon donor to the folate cycle, and allows the regeneration of methionine for SAM synthesis. Serine conversion to glycine by serine hydroxymethyl transferase (SHMT) generates 5,10-methylene-THF, which is subsequently reduced by methylene tetrahydrofolate reductase (MTHFR) to 5-methyl-THF within the folate cycle [40]. Methionine is regenerated by the donation of a methyl group to homocysteine by 5-methyl-THF. 

#### 3.1.1. DNA Methylation and Demethylation

The m5C methyltransferases (C-5 cytosine-specific DNA methylases) catalyze the methylation of the C-5 carbon of cytosines in CpG islands to produce C5-methylcytosine [46]. Given the known switch in global DNA methylation associated with gene promoters and gene bodies that takes place during embryonic development, a relevant question is whether DNA methylation patterns are subjected to systematic changes during specific processes that determine cell-fate, including EMT. Although no changes in bulk DNA methylation are observed during EMT induced by TGF-β1 [47,48], differentially-methylated regions (DMR) were found to be induced in Madin-Darby Canine Kidney (MDCK) and human breast cancer cells [48]. In the latter study, unmethylated sequences tended to become methylated, and affected both intergenic and intragenic regions, as well as promoters and gene bodies. As expected, methylation in promoters is associated with transcript downregulation, mainly affecting cell adhesion, catabolism and protein transport and methylation, while methylation in gene bodies was associated with transcript upregulation, which affected signaling pathways, developmental processes, wound-healing and cell differentiation. 

DNA can be indirectly demethylated through the deamination of 5mC by AID/APOBEC enzymes to give 5-hydroxymethyluracil [49], or through the oxidation of 5mC to 5-formylcytosine and 5-carboxylcytosine, catalyzed by ten-eleven translocation (TET) enzymes [50,51] (Figure 2). Activation-induced deaminase/apolipoprotein B mRNA editing enzymes, catalytic (AID/APOBEC) are zinc-dependent cytosine deaminases that function in antibody diversification and mRNA editing, with relatively weaker deamination-coupled demethylation activity on 5mC [49]. TET enzymes are dioxygenases that use Fe(2+) and α-KG as co-factors. The deaminated or oxidized adducts are repaired by thymine DNA glycosylase (TDG) and base excision repair [52,53,54]. The carboxyl and formyl groups of 5-formylcytosine and 5-carboxylcytosine can be enzymatically removed without the excision of the base [53].

#### 3.1.2. Histone Methylation and Demethylation

Histone methylation on lysine or arginine residues is catalyzed by lysine-specific (SET-domain or non-SET-domain) and arginine-specific histone methyltransferases (PRMTs) [55]. Histone methyl-ation at specific residues is associated with either the activation or the repression of transcription, depending on tethering “reader” proteins bearing domains that recognize specific histone modifications at specific lysines or arginines, which, in turn, recruit protein complexes that enable or repress transcription through chromatin remodeling [55]. Histone demethylation is catalyzed by two main classes of enzymes, flavin adenine dinucleotide (FAD)-dependent amine oxidases and Fe(2+) and α-α-KG-dependent hydroxylases. Both operate by the hydroxylation of a methyl group, followed by the dissociation of formaldehyde [56].

##### 3.1.2.1. FAD-Dependent Demethylation

Lysine-specific histone demethylase 1A (LSD1), also known as lysine (K)-specific demethylase 1A (KDM1A) [56], catalyzes the FAD-dependent demethylation of mono- and dimethyl groups, but not trimethyl groups, at histone H3K4 and H3K9, generating formaldehyde and H_2_O_2_. FAD is derived from riboflavin (vitamin B2), and serves as a coenzyme in many oxidative reactions including mitochondrial fatty acid β-oxidation and in the respiratory chain (Figure 2). The catalytic activity of LSD1 may be directly connected to the cellular metabolic state via the fluctuation of the FAD/FADH2 ratio depending on the FAD oxidation processes, such as fatty acid β-oxidation and the TCA cycle. LSD1 is an integral component of the Mi-2/nucleosome remodeling and deacetylase (NuRD) complex [56], that can function as a co-repressor or a co-activator, but dependent on the interaction with specific chromatin regulatory complexes. When forming complexes with co-repressors, such as SNAI1, LSD1 demethylates H3K4me1/2 and represses transcription [57,58]. When associated with the androgen or estrogen nuclear receptors, LSD1 demethylates H3K9me1/2 [59,60], and activates the transcription of pro-invasive and ECM remodeling genes. In contrast, it has also been found that LSD1 inhibits the invasion of breast cancer cells in vitro, and suppresses breast cancer metastatic potential in vivo [61].

##### 3.1.2.2. α-KG-Dependent Demethylation and Hydroxylation

The α-KG-dependent histone demethylases bear a Jumonji C domain (JmjC) [62]. Histone demethylation through the JmjC oxygenases occurs through a hydroxylation reaction, in which α-KG, oxygen, and Fe(2+) are used to produce succinate and CO_2_ (Figure 2). The JmjC histone demethylases regulate chromatin states through the removal of mono, di-, and tri-methylation marks upon specific lysine residues on histones. Different JmjC enzymes demethylate different methylated lysines on the histones, which imparts specific transcriptional outcomes (activation vs. repression) on each target gene [63]. 

Aside from JmjC demethylases, α-KG is a common substrate for several hydroxylases, including TET DNA hydroxylases [64] and prolyl hydroxylases, that regulate the stability of hypoxia-inducible factors (HIFs) [65]. Such hydroxylation reactions also require the co-factors Fe(2+) and vitamin C, as an electron donor that reduces ferric iron, Fe(3+), to ferrous iron, Fe(2+). This α-KG is an intermediary metabolite of the TCA cycle, produced by the isocitrate dehydrogenase (IDH)-catalyzed oxidative decarboxylation of isocitrate. This α-KG can also be produced in the reaction of glutamate and pyruvate, catalyzed by glutamate pyruvate transaminases (GPT1/2). Additionally, the reversible transfer of an amino group (NH_3_^+^) from glutamate to oxaloacetate, that has been catalyzed by glutamate oxaloacetate transaminases (GOT1/2) also results in the formation of α-KG and aspartate. Another non-TCA source of α-KG derives from glutaminolysis, in which glutamine transported into the cell is converted to glutamate and NH4+ in a deamination reaction catalyzed by glutaminases (GLS1/2). A second deamination reaction, catalyzed by mitochondrial glutamate dehydrogenase (GDH), reversibly converts glutamate to α-KG [66].

Importantly, succinate, fumarate, (D)-2-hydroxyglutarate (2-HG), and (L)-2-HG are α-KG structural analogs that competitively inhibit the hydroxylases which use α-KG as a substrate [65,67] (Figure 2). As a consequence, the relative concentrations of these metabolites are critical for histone and DNA methylation in the nucleus, and thus gene expression, as well as the regulation of HIF-1α levels. A compromise in SDH or FH activities leads to the accumulation of succinate or fumarate, respectively, causing the inhibition of DNA and histone demethylation [68,69]. Inactivating mutations in SDH are associated with pheochromocytomas and paragangliomas, sporadic renal cancer and gastrointestinal stromal tumors [69]. Accumulation of succinate causes epigenetic silencing of miR-200 and eventually EMT [70]. Similarly, loss-of-function FH mutations lead to hereditary leiomyomatosis and renal cell cancer (HLRCC), paragangliomas and pheochromocytomas [71,72,73]. Fumarate is accumulated in FH-deficient cells, inhibiting TET-dependent demethylation of miR200, which leads to its silencing and consequent EMT [68,70]. As a consequence of the competitive inhibition of α-KG by succinate and fumarate on proline hydroxylase activity, SDH or FH deficiencies promote the stabilization of HIF-1α in normoxic conditions, driving the expression of HIF target genes, including VEGF and GLUT1, that promote angiogenesis and glucose metabolism in renal and bile duct cancer cells [74].

IDH1 (cytosolic and peroxisomal) and IDH2 (mitochondrial) mutations occur frequently in a variety of human cancers, including malignant gliomas, AML, intrahepatic cholangiocarcinoma, chondrosarcoma, and thyroid carcinomas [75,76]. In addition, IDH2 mutations occur with high frequency in rare malignancies, such as angioimmunoblastic T cell lymphoma and solid papillary carcinoma with reverse polarity [77]. IDH-active site mutations confer a neomorphic activity that catalyzes the conversion of α-KG to D-2-hydroxyglutarate (D-2HG, or R-2HG) [76]. Under physiological conditions, cellular D-2HG accumulation is limited due to the actions of the endo-genous D-2HG dehydrogenase, which catalyzes the conversion of D-2HG to α-KG. Similar to high succinate or fumarate levels, high D-2HG levels competitively inhibit α-KG-dependent dioxygenases, causing the inhibition of JmjC and TET demethylases and histone and DNA hypermethylation, clinically associated with the increased methylation of patient tumor DNA in AML and gliomas, and in gliomas, a CpG island hypermethylator phenotype [76]. This is accompanied by an inhibition of normal differentiation processes and a promotion of tumorigenesis [78,79].

D-2HG levels have also been found elevated in breast cancer with wild-type IDH, driven by glutamine anaplerosis [80,81,82]. In colorectal cancer cells, D-2HG (but not its enantiomer L-2HG, produced from the reduction of α-KG by lactate dehydrogenase A, LDHA, or malate dehydrogenase, MDH) directly induces EMT in colorectal cancer cells by promoting H3K4me3 marks at the ZEB1 promoter and its transcription [83]. In this study, colorectal cancers with higher levels of D-2HG are associated with an increased frequency of distant metastasis, as well as an increased trend for a higher tumor stage. Further, hypoxia, independently of HIF, induces the LDH- and MDH-mediated production of the L-2HG enantiomer, which reinforces the hypoxic response, at least in part, through the stabilization of HIF-1α [84]. Accumulation of L-2HG, favored by acidic (low pH) conditions [85], slows glycolysis and mitochondrial respiration by reducing the rate of NAD+ regeneration [86], and promotes the same repressive chromatin marks that characterize the differentiation blockade of IDH-mutant malignancies. This provides a mechanistic link between hypoxic niches and stem-cell populations.

### 3.2. Acetylation

Protein acetylation is another major covalent modification directly linked to metabolite abundance that regulates gene programs as a result of histone acetylation and subsequent chromatin remodeling. The acetyl donor for acetylation reactions is acetyl-CoA, a central metabolic intermediate. The acetylation of histone lysines neutralizes the positive charges that govern the tight arrangement of nucleosomes and their interaction with DNA. As a result, chromatin becomes open to the access of bromodomain-containing proteins that dock at specific acetylated sites, and function as epigenetic readers or effectors, with critical roles in gene regulation. 

#### 3.2.1. Histone Acetylation and Deacetylation

The acetylation of the side-chain amino group of lysine residues on histones, and in some cases also other proteins, is mediated by histone acetyltransferases (HATs), or lysine acetyltransferases (KATs) [87]. HATs transfer the acetyl group from the acetyl-CoA cofactor to the ε nitrogen of a lysine side chain within the histones. In addition to charge neutralization and nucleosome remodeling, these histone modifications function as recognition sites for proteins bearing bromodomains (histone mark “readers”) that further reinforce the chromatin remodeling initiated by acetylation [88], leading to outcomes such as ATP-dependent H2A/H2B dimer eviction or complete nucleosome disassembly, and consequent transcriptional regulation.

Histone acetylation is balanced by deacetylation catalyzed by HDACs, of which there are three major families in mammals: Class I, class IIa, class IIb and class III or sirtuins [87]. Unlike the constitutively nuclear class I HDACs, class IIa HDACs do not display an intrinsic HDAC activity, which is acquired only when in complex with class I HDACs, once they enter the nucleus [87]. The cytoplasmic-nuclear shuttling of class IIa HDACs is regulated by LKB1/AMPK-mediated phosphorylation, which results in the retention of phosphorylated forms in the cytoplasm, while dephosphorylated forms shuttle to the nucleus, where they act as scaffolds for the class I HDACs that then exert transcription regulatory activities [89]. Therefore, the function of these HDACs is sensitive to nutrient availability and cellular energy status. The unrelated class III HDACs, or sirtuins [90], are NAD+-dependent deacylases localized in the nucleus (SIRT1, 3, 6 and 7), cytoplasm (SIRT2 and SIRT1) and the mitochondria (SIRT 3, 4 and 5). SIRT4 and SIRT6 also have ADP-ribosyl transferase activity, and SIRT5 demalonylase and desuccinylase activity.

#### 3.2.2. Regulation of Acetyl-CoA Pools

Global histone acetylation levels are sensitive to the availability of acetyl-CoA in the cell, which fluctuates in response to nutrient availability or metabolic reprogramming. In proliferating cells in culture, glucose fuels the majority of acetyl-CoA production used for acetylating histones. In mammalian cells, acetyl-CoA is produced within the mitochondria, the cytosol, and the nucleus [91]. Acetyl-CoA generated in mitochondria condenses with oxaloacetate to produce citrate, which is oxidized in the TCA cycle to provide ATP through OXPHOS. Citrate can be exported from mitochondria to the cytosol via the mitochondrial tricarboxylate transporter (SLC25A1), to regenerate acetyl-CoA and oxaloacetate by ATP-citrate lyase (ACLY). Because acetyl-CoA cannot be directly transported across mitochondrial membranes, citrate export and cleavage by ACLY is a major mechanism by which acetyl-CoA is generated outside of the mitochondria. Within the cytosol, acetyl-CoA is used in biosynthetic processes, including the synthesis of fatty acids and cholesterol. In addition to ACLY, another major source of acetyl-CoA outside of the mitochondria is ACSS2, which is localized to the cytosol and nucleus. ACSS2 is involved in the capture and use of exogenous acetate, as well as in the recycling of acetate produced by histone deacetylase (HDAC) reactions [92]. Additionally, the PDC can translocate to the nucleus under certain conditions, such as mitochondrial stress, where it contributes to provide acetyl-CoA for histone acetylation [93].

Since glucose is the preferred source of acetyl-CoA in proliferating cells, glucose limitation or glycolytic inhibition suppresses both acetyl-CoA and histone acetylation levels [94]. However, some cells can use carbon sources other than glucose to produce acetyl-CoA, such as exogenous acetate, in particular under metabolic stress conditions in hypoxia or fasting [95]. Acetate can be converted to acetyl-CoA by ACSS2, which is translocated to the nucleus in low oxygen and glucose conditions [96,97], thus mediating the recycling of the acetate produced by the HDAC reactions, instead of incorporating exogenous acetate. Therefore, nuclear ACSS2 may primarily rely on a locally-generated acetate pool for histone acetylation, while cytosolic ACSS2 promotes the use of exogenous acetate for lipid synthesis.

#### 3.2.3. Regulation of NAD+/NADH Pools

NAD+ is a hydride-transfer acceptor, serving a wide variety of metabolic transformations, as it interconverts between its oxidized form (NAD+) and its reduced form (NADH) [98]. NAD+ directly participates in compartment-specific central metabolic pathways, such as glycolysis, PDH, the TCA cycle, fatty acid oxidation, amino acid oxidation and OXPHOS. In glycolysis, NAD+ is converted to NADH in the glyceraldehyde-3-phosphate dehydrogenase step. This process occurs in reverse for gluconeogenesis in the liver. NAD+ is also a precursor substrate for NADP+ and NADPH, which participate in biosynthesis and reactive oxygen detoxification. NADPH is a key reducing substrate to convert oxidized glutathione to reduced glutathione, a key protectant for cells to resist the toxicity of ROS.

In the nucleus, apart from its role in histone deacetylation, NAD+ is used by PARP, with essential functions in DNA damage repair, as well as by cyclic ADP-ribose synthases [99]. NAD+ is actively consumed during these enzymatic processes, serving as the donor of ADP-ribose in the reaction. The majority of PARP activity is distributed between PARP-1 and PARP-2. As the Km of PARP-1 for NAD+ is below its nuclear concentration, it is unlikely that the activity of PARP-1 is significantly affected by the fluctuations of NAD+. However, PARP-1 activity can reduce the effective concentration of the NAD+ available for other enzymes. As such, consumption of NAD+ by constitutive activation of PARP-1 hampers SIRT1 activity [99,100]. The PARP-2 dissociation constant for NAD+ is within the range of the physiological changes in NAD+ concentration (Km = 130 μM), and thus PARP-2 can directly compete with SIRT1 for NAD+ [101].

To summarize, the regulation of the expression of specific genes and entire gene programs by histone acetylation is orchestrated by metabolic inputs at multiple levels, including the availability of nuclear acetyl-CoA and the expression, activity and localization of HATs, bromodomain proteins and histone deacetylases, which in turn are also regulated by cellular metabolic states. Many of the genes thus regulated by metabolic inputs control the expression of proteins pertaining to the same or separate metabolic circuits, forming complex regulatory loops that enable a rapid fine-tuning of cell metabolic adaptations in response to a multiplicity of scenarios, from shifts in nutrient availability to environmental or oncogenic stress. Thus it is no surprise that changes in global or locus-specific histone acetylation have a significant impact on epithelial-mesenchymal plasticity, and conversely, these shifts utilize changes in histone acetylation.

In early embryonic development, mesoderm specification is accompanied by a downregulation of class I HDACs, and an induction of global histone acetylation by a treatment with the HDAC inhibitor trichostatin A (TSA) drives differentiation to the mesodermal lineage [102]. Consistently, treatment of epithelial tumor cells with TSA or other HDAC inhibitors, such as suberoylanilide hydroxamic acid (SAHA) or valproic acid (VPA), can induce EMT in certain cancer cells, with an upregulation of EMT factors such as ZEB1, ZEB2 or SLUG (SNAI2) [103,104,105,106,107]. This might explain the disappointing outcomes in the clinical trials of HDAC-targeting monotherapies for solid tumors [108]. However, HDACs are required to initiate or maintain EMT in other circumstances [109,110,111,112,113,114]. 

A possible explanation for these discrepant observations is the timing, dosing or the potency of HDAC inhibitor administration. As global histone acetylation generally enables open chromatin conformations associated with active transcription, the inhibition of HDACs or an abnormal accumulation of acetyl-CoA, as reported for hepatocellular carcinoma [115], would favor the expression of transcription factors necessary to initiate EMT in response to appropriate signals [116,117]. These factors tend to be transcriptional repressors for epithelial target genes, such as *CDH1*, for which function they recruit HDACs [118,119,120,121,122,123]. During EMT, the early repression of epithelial genes is later followed by a transcriptional activation of mesenchymal genes [124], which requires an open chromatin conformation enabled by the gain of histone acetylation. At least for *SNAI1*, the late wave of the transcriptional activation of mesenchymal genes is not associated with the binding of the EMT factor to their promoters [125]. These observations suggest that, in its early stages, EMT is established on cycles of transient histone acetylation and deacetylation at specific genes. The progressive stabilization of EMT is achieved through the deposition of any marks of stable chromatin repression [124] and, eventually, heritable DNA methylation of epithelial genes and regulators, such as the microRNA-200 family [125,126]. Cells that are in plastic, “intermediate” EMT states display bivalent marks associated with the promoters of EMT factors and effectors [124,127].

The effects of metabolism on the putative regulatory roles of the NAD+-dependent sirtuins on epithelial-mesenchymal plasticity is even less well understood than for class I and II HDACs [128]. Many reports generally provide evidences that one or more sirtuins favor EMT [129,130,131,132,133]. The ultimate mechanism of this function may be the recruitment of SIRT1 to the *CDH1* promoter by ZEB1 to the deacetylate histone H3 suppressing *E-cadherin* transcription [133,134,135], with the possible participation of other epigenetic regulators, such as MPP8, a methyl-H3K9 binding protein [136]. If this is the case, a depletion of NAD+, which occurs upon extensive DNA damage and an activation of PPAR with the consumption of NAD+, or during hypoxia, would be expected to blunt the activity of sirtuins and to impair EMT.

Figure 3 summarizes the descriptions provided in this section of some of the major directions that metabolic networks take to enable the epigenetic modifications necessary for the execution of the transcriptional programs orchestrating epithelial-mesenchymal plasticity. 

## 4. EMT, Metabolism and Hypoxia

Hypoxia, nutrient starvation and lactate acidosis can each regulate gene expression at the transcriptional and posttranscriptional levels in vitro. Intratumoral hypoxia occurs when the partial pressure of O_2_ is <5%, as tumor growth outpaces neoangiogenesis, generating heterogeneous O_2_ gradients throughout the tumor. Tumor hypoxia promotes chemoresistance and radiation resistance [137]. Both the stabilization and activation of HIF-1α promote adaptation to hypoxic stress by modulating tumor cell metabolism, survival, angiogenesis, migration, invasion and metastasis. 

High HIF-1α expression is associated with poor prognosis in many cancers [138]. Hypoxia induces ameboid motility [139] and invasive phenotypes, mediated by HIF-1α in multiple cancer types [140,141]. Interestingly, the enhanced motility and invasion elicited by hypoxia is not necessarily accompanied with an increased proliferation in ovarian cancer cells [142].

In normoxia, HIF-1α is a substrate of hydroxylation on specific prolines, catalyzed by oxygen-sensing prolyl hydroxylase domain (PHD) enzymes. Hydroxyprolyl-modified HIF-1α is recognized by the Cul2 subunit of the VHL E3 ubiquitin ligase, which mediates the polyubiquitylation and proteasome-mediated degradation of HIF-1α [143]. Low O_2_ levels are accompanied with low PHD activity, and as a consequence, HIF-1α is stabilized during hypoxia. Loss-of-function mutations that affect the VHL ubiquitin ligase function, occurring in the von Hippel-Lindau disease [144], lead to an aberrant accumulation of HIF-1α. Since PHDs are αKG-dependent oxygenases [145], low levels of α-KG or high levels of its structural analogs and competitors succinate or fumarate (see Section 3.1.2.2) inhibit PHD enzymatic activity, causing the stabilization and accumulation of HIF-1α and leading to a state of pseudohypoxia.

As a transcription factor, HIF-1α activates the expression of the EMT factors *SNAI1* [146], *ZEB1/2* [147], *TCF3* [148,149] and *TWIST1* [150] to repress *E-cadherin* expression. It also elicits the expression of lysyl oxidase [151] and matrix metalloproteases for extracellular matrix remodeling, as well as angiogenic factors to promote the vascularization of the hypoxic areas and erythropoietin to boost red blood cell production in the bone marrow [138].

The major effects of HIF-1α activation on metabolism are to (1) stimulate glycolytic energy production by promoting the expression of the glucose transporter GLUT1 (SLC2A1) and glycolytic enzymes (such as hexokinase 1/2 (HK1/2), PFK1, PFKFB3 and aldolase); and (2) to downregulate mitochondrial OXPHOS by promoting the expression pyruvate dehydrogenase kinase 1 (PDK1) [152] and the MYC inhibitor MAX interactor 1 (MXI1) in renal cell carcinoma cells [153]. The combination of blunted mitochondrial function and high glycolytic activity associated with hypoxia and HIF-1α activity leads to the accumulation of cytoplasmic pyruvate and NADH. In order to dispose of these compounds, lactate dehydrogenase A (LDHA) is induced by HIF-1α and catalyzes the conversion of pyruvate and NADH to lactate and NAD+ [153], after which lactate is exported to the extracellular milieu through the HIF-inducible plasma membrane monocarboxylate transporter 4 (MCT4, SLC16A4) [154]. Thus, a collateral effect of these processes is the acidosis of the extracellular environment surrounding the tumor cells under hypoxia.

Acidosis is a characteristic feature of the tumor microenvironment that directly regulates tumor cell invasion by affecting immune cell function, clonal cell evolution and drug resistance [155]. Unlike normal cells, cancer cells can adapt to survive in low pH (acidic) environments through increased glycolytic activity and an expression of proton transporters that normalize intracellular pH [155]. Acidosis-driven adaptation also triggers the emergence of aggressive tumor cell subpopulations that exhibit increased invasion, proliferation and drug resistance [155]. Low extracellular pH induces increased histone deacetylation, thereby influencing the expression of stress-responsive genes and contributing to the normalization of intracellular pH through the enhanced release of acetate anions that are co-exported with protons through monocarboxylate transporters [156,157]. Low pH areas of these tumors are not necessarily restricted to hypoxic areas, and are enriched for cells that are invasive and proliferative. Acidic conditions induce a reversible transcriptomic rewiring, independent of lactate, involving RNA splicing and being enriched for the targets of RNA binding proteins with specificity for AU-rich motifs, including CD44 [158]. CD44 is a hyaluronan-binding receptor that mediates cell invasiveness and motility [159], expressed as two major isoforms associated with either epithelial or mesenchymal gene programs, and regulated by the alternative splicing factors ESRP1/2, that promote an epithelial program [160] or QKI and ROBFOX1/2, that promote a mesenchymal program [161,162,163].

## 5. EMT, Metabolism and Drug Resistance 

Numerous experimental reports and studies with patient cohorts have found a significant association of EMT or mesenchymal traits of tumors with acquired resistance to chemotherapy, targeted drugs and immunotherapy [164,165,166]. No unifying theme has emerged that explains the specific mechanisms co-opted by EMT as a path to drug resistance. In one case, early adaptive resistance of lung cancer cells to the EGFR inhibitor erlotinib led to a metabolically quiescent state, albeit with increased glutamine addition and a survival attributed to an enhanced expression of BCL-2 and BCL-xL [167]. Resistance to the mitotic drug docetaxel was found to induce EMT in prostate cancer cells, accompanied with a more efficient respiratory phenotype, a utilization of glucose and glutamine and the production of lactate [168]. Short-term treatment of colorectal cancer cells with docetaxel induced an EMT phenotype in the surviving cells, which were more dependent on OXPHOS than sensitive cells [169]. Studies by our laboratory comparing the metabolic dependencies and vulnerabilities of clonal prostate cancer cell lines displaying either epithelial or marked mesenchymal phenotypes, indicated the establishment of a Warburg effect in epithelial cells with high lactate production and a strong reliance on glutaminolysis for the anaplerotic feeding of the TCA cycle. This rendered epithelial CSCs highly sensitive to glutaminase inhibitors, while mesenchymal cells more efficiently funneled glucose to the TCA cycle, displayed stronger OXPHOS dependency, and were more sensitive to mitochondrial complex I and III inhibitors [170].

When cells display a Warburg effect with an enhanced production of lactate, as observed in cells in intermediate EMT states concomitant with CSC features, the resulting acidic extracellular environment can promote drug resistance through several mechanisms, including the so-called “ion trapping” mechanism, by which molecules that are weak bases (e.g., anthracyclines, anthraquinones or vinca alkaloids) are protonated at an acidic pH, which impairs their diffusion through plasma membranes [171]. On the other hand, weak acids (e.g., chorambucil, cyclophosphamide, 5-FU) are not ionized at an acidic pH, and can more readily traverse membranes, reaching the slightly alkaline intracellular milieu, where they become negatively-charged, and accumulate for more efficacious cytotoxic effects [172]. Resistance to immunotherapy and immunosuppression is also tightly linked to tumor-generated metabolic microenvironments, including nutrient exhaustion (including glucose, glutamine or tryptophan) [173], hypoxia, or also enhanced acidosis by accumulation of lactate [174]. A more direct connection between EMT and tumor immune escape is provided by the demonstration that the immune checkpoint ligand PD-L1 is a target of miR-200 [175]. As such, ZEB1 expression downregulates miR-200 and enhances the expression of PD-L1 on tumor cells undergoing EMT. In turn, ligand-engaged PD-L1 can induce or reinforce EMT, as exemplified in renal cell carcinoma cells [176], thus establishing a positive feedback loop that potentiates acidic, inflammatory and immunosuppressive tumor microenvironments, that are also predicted to result in enhanced resistance to other drugs.

Therefore, differential glycolytic and mitochondrial efficiencies associated with epithelial or mesenchymal plasticity confer differential metabolic dependencies which can be exploited therapeutically in combinatorial schemes intended at overcoming drug resistance strategies adopted by cancer cells [177], including undergoing an EMT.

## 6. Concluding Remarks

From their earliest stages in development, multicellular organisms engage in asymmetric division followed by cell-cell communication in order to generate heterogeneity and specialization. Each stage in cell fate determination is exquisitely orchestrated by specific regulators that coordinate intricate transcriptional, signaling and metabolic networks that ultimately define cell identity and tissue and organ functions. Epithelial cells that enter a given differentiation lineage maintain degrees of plasticity before they acquire stable specialized features, and as such, they may exit or revert their committed paths, notably through EMT. In cancer, transformed cells endowed with phenotypic plasticity co-opt these same mechanisms as they evolve and adapt in response to environmental challenges, yielding intratumoral heterogeneity, which significantly impacts the biology and management of cancers.

Epithelial-mesenchymal plasticity is so deeply intertwined with metabolic reprogramming that the two processes may no longer be considered separately, in physiological or pathological scenarios. As such, signaling cues that drive EMT must concomitantly induce an appropriate reprogramming of metabolic networks in order to meet the requirements of the new cellular state, while conversely, endogenous or exogenous shifts in metabolic balances can drive EMT in their own right. This perspective opens the prospect that epithelial-mesenchymal plasticity may be modulated, in both clinical and experimental settings, through appropriate metabolic interventions. As discussed above, the inhibition of metabolic enzymes such as ALDOA, GAPDH or FASN can prevent or revert EMT in cell models, as can the neutralization of acidic tumor environments. The recent observations that mesenchymal properties require fused mitochondria endowed with efficient TCA and OXHPOS [10], while prostate CSCs rely on mitochondrial fission [178], afford the prediction that interventions directly aimed at mitochondrial function or dynamics will produce differential effects on cell plasticity, and thus are worth exploring as antineoplastic strategies.

Other metabolic interventions with the potential of impacting EMT and, consequently, tumor heterogeneity and drug resistance, include those that shift the intracellular balance of metabolites with direct effects on epigenetic marks and gene regulation. In this regard, an interesting recent study [179] has shown the feasibility of dietary methionine restriction as a strategy to target one-carbon metabolism, which modulated tumor growth and conferred tumor chemosensitivity to conventional drugs, thus opening the door to novel mechanistically-based dietary interventions designed at targeting specific metabolic pathways deranged in a given tumor type.

In summary, a deeper understanding of the multiple-level links between metabolism, epigenetics and epithelial-mesenchymal transition and their intimate interconnections should lead to the rational discovery of primary vulnerabilities as well as secondary vulnerabilities that emerge in response to first-line therapies, thus paving the way to propose combinatorial therapeutic strategies to combat advanced cancers and pharmacological or immunological resistance.

## Figures and Tables

**Figure 1 jcm-08-00967-f001:**
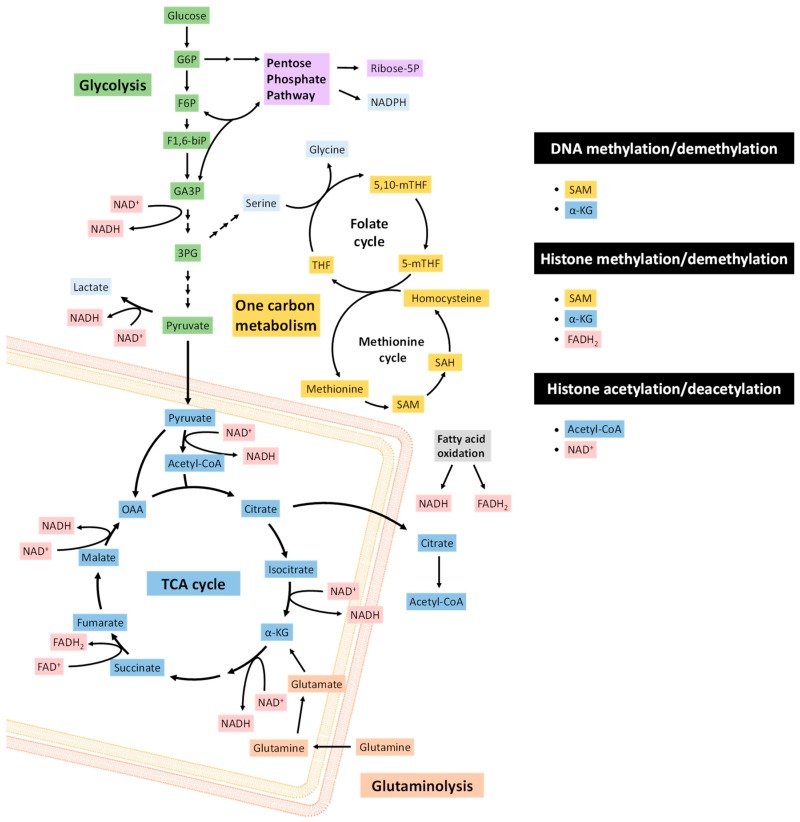
Metabolic requirements of protein and DNA methylation and protein acetylation. S-adenosyl-methionine (SAM) acts as a methyl donor for histone and DNA methylation. SAM is produced through one-carbon metabolism. Demethylation can require the tricarboxylic acid (TCA) cycle intermediate α-KG as a co-factor, and can be inhibited by other TCA cycle products: Succinate, fumarate or 2-hydroxyglutarate (2-HG). Histone demethylation by lysine-specific histone deacetylase 1 (LSD1) can also require the flavin adenine dinucleotide reduced form (FADH2), whose pools are dependent upon fatty acid oxidation and the TCA cycle. Histone acetylation requires acetyl-CoA, obtained from citrate or fatty acid oxidation in the TCA cycle. Histone deacetylation by sirtuin (SIRT) histone deacetylases is NAD+-dependent. NADH pools derive from multiple metabolic pathways, including glycolysis, pyruvate oxidation by pyruvate dehydrogenase (PDH), the TCA cycle, fatty acid oxidation, amino acid oxidation and OXPHOS. The metabolites involved in different pathways are shaded in different colors: Glycolysis (green), one carbon metabolism (yellow), pentose phosphate pathway (purple), TCA cycle (blue), NADH and FADH2 (pink) and glutaminolysis (orange).

**Figure 2 jcm-08-00967-f002:**
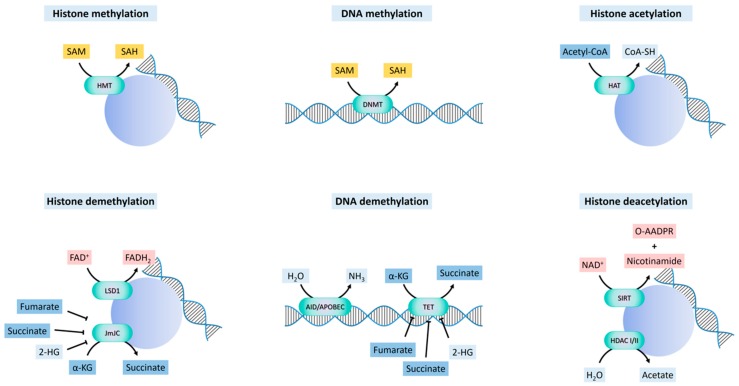
Mechanisms of histone and DNA methylation/demethylation and histone acetylation/deacetylation. S-adenosyl-methionine (SAM) acts as a methyl donor for histone and DNA methylation, yielding S-adenosyl homocysteine (SAH). DNA methylation is performed by DNA methyl transferases (DNMT), whereas histone methylation is performed by histone methyltransferases (HMT). Histone demethylation can be mediated by flavin adenine dinucleotide (FAD)-dependent lysine-specific demethylase 1 (LSD1); or by Jumonji C domain (JmjC) demethylases, that use α-ketoglutarate (α-KG) as a substrate, yielding succinate. JmjC demethylases can also be inhibited by other TCA cycle products: Succinate, fumarate or 2-hydroxyglutarate (2-HG). DNA demethylation can occur through the activation-induced deaminase/apolipoprotein B mRNA editing enzyme, catalytic (AID/APOBEC), or by ten-eleven translocation (TET) enzymes. TET dioxygenases use Fe(2+) and α-KG as co-factors, and their activity can also be inhibited by fumarate, succinate and 2-HG. Histone acetylation by histone acetyl-transferases (HAT) requires acetyl-CoA. Histone deacetylation can occur by class I and II histone deacetylases (HDAC I/II), and by class III histone deacetylases, also termed as sirtuins (SIRT). Sirtuins use nicotinamide adenine dinucleotide (NAD+), which is converted into nicotinamide and O-acetyl-ADP-ribose (O-AADPR).

**Figure 3 jcm-08-00967-f003:**
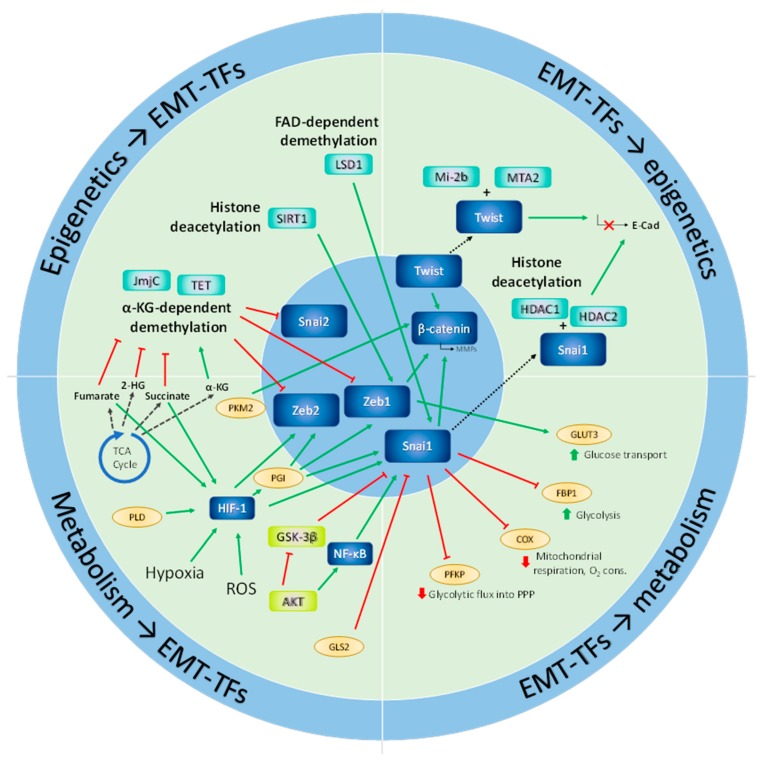
Relevant examples of crosstalk between metabolism, epithelial-mesenchymal plasticity and epigenetics. Epithelial-mesenchymal transition (EMT)-transcription factors (EMT-TFs) are regulated by histone and DNA methylation and by histone acetylation. These processes are, in turn, modulated by the intracellular levels of the metabolic products of the tricarboxylic acid (TCA) cycle. EMT-TFs also cooperate with histone deacetylases to repress the expression of cell adhesion molecules that modulate the EMT phenotype. Metabolic cues such as hypoxia, oxidative stress or nutrient availability can activate EMT-TFs through various signaling axes, such as HIF-1α, Akt, GSK-3β or NF-κB. Conversely, the expression of EMT-TFs can be directly modulated by metabolic enzymes, such as phosphoglucoisomerase (PGI) or pyruvate kinase isoform M2 (PKM2). In turn, EMT-TFs regulate central metabolic pathways, by activating or repressing the transcription of metabolic enzymes and metabolite transporters, such as phosphofructokinase platelet (PFKP), cytochrome c oxidase (COX), fructose-1,6-bisphophatase 1 (FBP1), or glucose transporter 3 (GLUT3).

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
