# Peer review of "Metabolic Plasticity and Epithelial-Mesenchymal Transition"

_jcm, 2019, doi:10.3390/jcm8070967_

Reviewer 1 Report

The manuscript entitled “Metabolic plasticity and epithelial-mesenchymal transitions” is a very interesting review article; as the authors have reported numerous works in the literature, presenting the EMT phenomenon from epigenetic and metabolic point of view. The authors have organized several experimental results very well, providing a complete picture of the state of the art.

I very appreciate this manuscript. I only suggest at authors to optimize the quality of figures and to improve the sentences, mainly in the Introduction section, in order to be easier to understand for readers.

Author Response

We are thankful for your kind words. We have intended to upgrade the overall quality of the manuscript and figures to meet all the reviewers' requirements.

Reviewer 2 Report

This review by Thompson and colleagues examines the metabolic features of cells undergoing EMT and in particular epigenetic modifications. The review is comprehensive in the area but has issues which need to be addressed.

This manuscript is very difficult to read. The style of writing is confusing at times and difficult to follow. Eg. "The development of tissue and organ specialization in multicellular organisms requires multiple steps of binary decisions made by precursor cells in their process to produce differentiated  progeny.", "resulting in a zero-sum scenario", "From a metabolic budget perspective, the continuum...", "through the characterization of the distribution of metabolic fluxes of cancer cells when subjected to particular perturbations" etc.

The review is very long and could possibly be cut by 25%. It would be far more useful to have more diagrams rather than heavy text.

A lot of detail is presented but the point of the review is lost.

Author Response

We appreciate all the remarks made by the reviewer. We have attempted to simplify our style with shorter and clearer sentences and we have shortened the manuscript where possible.

Reviewer 3 Report

1) In “This suggests a balance between motility and proliferation [4, 5] that may depend on the relative availability of common resources  that that can be spent on either motility or proliferation, resulting in a zero-sum scenario from a metabolic standpoint”, please, remove one that.

2) I suggest to rewrite this sentence: “It is also indicative of the existence of a range of EMT states [6], with the corollary that not all EMTs (or METs) are equal: while “extreme” EMT can lead to stable  mesenchymal phenotypes prone to adopt pre-senescent states [5], “partial” EMT endows cells with features shared with stem cells, including self-renewal or survival under stress and in non-adherent  growth conditions [7, 8].”

3) This paragraph is a repetition of sentences previously reported “As such, a continuum spectrum of EMT-MET transition states 53 may be the reflection of a dynamic balance between processes that lead to loss of cell-cell adhesion 54 and gain of mobility, on the one hand, and those favoring cell growth and proliferation, on the other”

4) This sentence is confusing in its meaning. “This would also imply that “pure” fate-transition processes would not be easy to uncouple from 56 each other in “partial” or “intermediate” EMT states, but they may be separately addressed at either 57 end of the transition spectrum, i.e., when displaying stable, non-overlapping epithelial vs. 58 mesenchymal phenotypes “

5) At the end of the introduction I suggest to indicate the aim of the review.

6) Please, summarize this part avoiding the repetitions: Metabolism: energy and building blocks.

7) Above all, I suggest to rewrite the sentences “Phenotypic plasticity is closely associated with  metabolic plasticity, as both are determined by, and impact, transcriptional networks and, as such, alternative phenotypes driven by alternative sets of transcription factors may be expected to be  associated with alternative metabolic states. In this regard, and potentially relevant to the acquisition  of alternative epithelial and mesenchymal gene programs, two major carbon flux pathways, glycolysis and one-carbon metabolism, exhibit a mutually inverse correlation”

8) I suggest to rewrite the sentence “Understanding metabolic plasticity through the characterization of the distribution of metabolic   fluxes of cancer cells when subjected to particular perturbations (i.e. chemotherapeutic regime,   nutrient starvation, hypoxia, etc.) can expose their vulnerabilities, and thus provide novel   therapeutic strategies.

9) Ii is s not appropriate to put this sentence on lane 123-4. “It would be of interest to determine whether these discrepancies in metabolic  rewiring are associated with the degree of EMT achieved in each of these experimental models”.

10) For each paragraphs it is very important to insert the type of cancer cells the authors refer to.

11) Please, rewrite the following sentences:In addition to ATP deficiency, elevated ROS levels can further contribute to the  activation of anoikis, since ROS accumulation inhibits ATP production [32]. Therefore, many of the 148 adaptations exhibited by cancer cells to evade anoikis may also be dedicated to scavenging ROS or 149 diminishing their generation. One of these adaptations may be the reinforcement of a 150 highly-glycolytic phenotype, relying on glucose to generate ATP, which can decrease cellular ROS 151 levels both by diminished mitochondrial metabolism (Crabtree effect; [33]) and increased NADPH 152 production capacity through the pentose phosphate pathway (PPP) [34]”

12) I suggest to rewrite the sentence. “Metabolic and EMT signaling cascades display a remarkable level of overlapping reflective of  their coupling for conferring a cancer cell phenotype”.

13) Please, explain in details what is the link between the metabolic and the EMT phenotypes for  GSK3.

14) In Epigenetic determinants of epithelial-mesenchymal plasticity section, the authors refer to transcription factors and immediately after they talk about enzymes that carry out epigenetic modifications. Please, add information about transcription factors. Moreover, the authors should describe in details the different kind of tumor cells they referred to in this section.

15) Figure 1 is completely unnecessary in this form. The authors should remade this in a more specific way focus their attention of enzymes that carry out epigenetic modifications. Please, avoid metabolism mistakes (reactions arrows) and extra-information, explain the meaning of the colours.

16) Concluding remarks are inappropriate as they do not summarize the whole review. The point of view of the authors is necessary.

Author Response

1) In “This suggests a balance between motility and proliferation [4, 5] that may depend on the relative availability of common resources that that can be spent on either motility or proliferation, resulting in a zero-sum scenario from a metabolic standpoint”, please, remove one that.

We have removed one “that” [line 43]

2) I suggest to rewrite this sentence: “It is also indicative of the existence of a range of EMT states [6], with the corollary that not all EMTs (or METs) are equal: while “extreme” EMT can lead to stable  mesenchymal phenotypes prone to adopt pre-senescent states [5], “partial” EMT endows cells with features shared with stem cells, including self-renewal or survival under stress and in non-adherent  growth conditions [7, 8].”

This sentence has been rewritten in the new version of the manuscript [lines 45-48]

3) This paragraph is a repetition of sentences previously reported “As such, a continuum spectrum of EMT-MET transition states 53 may be the reflection of a dynamic balance between processes that lead to loss of cell-cell adhesion 54 and gain of mobility, on the one hand, and those favoring cell growth and proliferation, on the other”

We have rewritten the sentence into a shorter version: “As already mentioned, the continuum spectrum of EMT-MET transition states mirrors the dynamic balance between motility and proliferation.” [lines 51-55].

4) This sentence is confusing in its meaning. “This would also imply that “pure” fate-transition processes would not be easy to uncouple from 56 each other in “partial” or “intermediate” EMT states, but they may be separately addressed at either 57 end of the transition spectrum, i.e., when displaying stable, non-overlapping epithelial vs. 58 mesenchymal phenotypes “

We have  rewritten this sentence in a clearer manner [lines 51-55].

5) At the end of the introduction I suggest to indicate the aim of the review.

We are thankful for this appreciation. We have added a paragraph indicating the aim of the review at the end of the introduction [lines 56-58].

6) Please, summarize this part avoiding the repetitions: Metabolism: energy and building blocks.

We have eliminated redundant expressions from the first paragraph in this section [lines 70-82].

7) Above all, I suggest to rewrite the sentences “Phenotypic plasticity is closely associated with  metabolic plasticity, as both are determined by, and impact, transcriptional networks and, as such, alternative phenotypes driven by alternative sets of transcription factors may be expected to be  associated with alternative metabolic states. In this regard, and potentially relevant to the acquisition  of alternative epithelial and mesenchymal gene programs, two major carbon flux pathways, glycolysis and one-carbon metabolism, exhibit a mutually inverse correlation”

These sentences have been rewritten [lineas 67-69].

8) I suggest to rewrite the sentence “Understanding metabolic plasticity through the characterization of the distribution of metabolic   fluxes of cancer cells when subjected to particular perturbations (i.e. chemotherapeutic regime,   nutrient starvation, hypoxia, etc.) can expose their vulnerabilities, and thus provide novel   therapeutic strategies.

We have rewritten this sentence in the revised version.

9) It is not appropriate to put this sentence on lane 123-4. “It would be of interest to determine whether these discrepancies in metabolic rewiring are associated with the degree of EMT achieved in each of these experimental models”.

This sentence has been removed in the revised manuscript.

10) For each paragraphs it is very important to insert the type of cancer cells the authors refer to.

We have indicated the type of cell models used in most of the referenced literature in the corrected version of the manuscript. If this clarification is missing, findings were reported in multiple types of cell models.

11) Please, rewrite the following sentences: “In addition to ATP deficiency, elevated ROS levels can further contribute to the activation of anoikis, since ROS accumulation inhibits ATP production [32]. Therefore, many of the 148 adaptations exhibited by cancer cells to evade anoikis may also be dedicated to scavenging ROS or 149 diminishing their generation. One of these adaptations may be the reinforcement of a 150 highly-glycolytic phenotype, relying on glucose to generate ATP, which can decrease cellular ROS 151 levels both by diminished mitochondrial metabolism (Crabtree effect; [33]) and increased NADPH 152 production capacity through the pentose phosphate pathway (PPP) [34]”

These sentences have been rewritten. We hope to have achieved improved clarity on this idea [lineas 140-146].

12) I suggest to rewrite the sentence. “Metabolic and EMT signaling cascades display a remarkable level of overlapping reflective of their coupling for conferring a cancer cell phenotype”.

We have rewritten this sentence for improved clarity [lines 157-166].

13) Please, explain in details what is the link between the metabolic and the EMT phenotypes for  GSK3.

The link between AKT, GSK3 and SNAI1 has been further nuanced. Also, a reference for this fragment was missing and it has now been added in the revised version [lines 157-166].

14) In Epigenetic determinants of epithelial-mesenchymal plasticity section, the authors refer to transcription factors and immediately after they talk about enzymes that carry out epigenetic modifications. Please, add information about transcription factors. Moreover, the authors should describe in details the different kind of tumor cells they referred to in this section.

For better context, we have named several EMT TFs and microRNAs [line 174]. We believe that a full description of such factors and their mechanisms of action is outside of the scope of this review.

We have modified all figures so as achieve better adaptations with the descriptions in the text and to improve illustration of relevant enzymatic reactions.

The tumor cell models used in the referenced literature are now described in the text.

15) Figure 1 is completely unnecessary in this form. The authors should remade this in a more specific way focus their attention of enzymes that carry out epigenetic modifications. Please, avoid metabolism mistakes (reactions arrows) and extra-information, explain the meaning of the colours.

We have modified all figures so as achieve better adaptations with the descriptions in the text and to improve illustration of relevant enzymatic reactions.

16) Concluding remarks are inappropriate as they do not summarize the whole review. The point of view of the authors is necessary.

We have fully rewritten the concluding remarks, expressing not a summary of the review but rather relevant perspectives by the authors on the significance of the interconnections between metabolism and EMT, with emphasis on therapeutic approaches tackling EMT plasticity with metabolism modifiers [lines 760-795].

Round  2

Reviewer 2 Report

This manuscript is much improved, especially the beginning of the manuscript, however it is still hard to read and tedious in sections. Can be made more succinct. Some more careful editing and reduction in length is required. Also more citations are necessary in some areas of the manuscript.

Author Response

This manuscript is much improved, especially the beginning of the manuscript, however it is still hard to read and tedious in sections. Can be made more succinct. Some more careful editing and reduction in length is required. Also more citations are necessary in some areas of the manuscript.

We thank the reviewer for the positive comments.

- We have removed several paragraphs (lines 60-69, 83-95, 468-472, 477-479, 484-487, 722-726, 744-745 in the previous version of the manuscript). 

- We have added new references (57, 67, 68, 84, 125-127, 143-146) as deemed pertinent to the text descriptions.

- We have edtited the entire manuscript in an attempt to improve the phrasing where it appeared to be suboptimal.

We hope to have achieved the task of improving the readability of our manuscript, in accordance with the reviewer's indications.